# Multiple Myeloma: Genetic and Epigenetic Biomarkers with Clinical Potential

**DOI:** 10.3390/ijms252413404

**Published:** 2024-12-13

**Authors:** Yuliya A. Veryaskina, Sergei E. Titov, Natalia V. Skvortsova, Igor B. Kovynev, Oksana V. Antonenko, Sergei A. Demakov, Pavel S. Demenkov, Tatiana I. Pospelova, Mikhail K. Ivanov, Igor F. Zhimulev

**Affiliations:** 1Laboratory of Molecular Genetics, Department of the Structure and Function of Chromosomes, Institute of Molecular and Cellular Biology, Siberian Branch of the Russian Academy of Sciences, Novosibirsk 630090, Russia; titovse78@gmail.com (S.E.T.); ovant@mcb.nsc.ru (O.V.A.); demakov@mcb.nsc.ru (S.A.D.); zhimulev@mcb.nsc.ru (I.F.Z.); 2Laboratory of Gene Engineering, Institute of Cytology and Genetics, Siberian Branch of the Russian Academy of Sciences, Novosibirsk 630090, Russia; 3AO Vector-Best, Novosibirsk 630117, Russia; ivanovmk@vector-best.ru; 4Department of Therapy, Hematology and Transfusiology, Novosibirsk State Medical University, Novosibirsk 630091, Russia; nata_sk78@mail.ru (N.V.S.); kovin_gem@mail.ru (I.B.K.); depart04@mail.ru (T.I.P.); 5Laboratory of Computer Proteomics, Institute of Cytology and Genetics, Siberian Branch of the Russian Academy of Sciences, Novosibirsk 630090, Russia; demps@bionet.nsc.ru

**Keywords:** microRNA, oncogenes, tumor suppressors, multiple myeloma, prognostic biomarkers

## Abstract

Multiple myeloma (MM) is characterized by the uncontrolled proliferation of monoclonal plasma cells and accounts for approximately 10% of all hematologic malignancies. The clinical outcomes of MM can exhibit considerable variability. Variability in both the genetic and epigenetic characteristics of MM undeniably contributes to tumor dynamics. The aim of the present study was to identify biomarkers with the potential to improve the accuracy of prognosis assessment in MM. Initially, miRNA sequencing was conducted on bone marrow (BM) samples from patients with MM. Subsequently, the expression levels of 27 microRNAs (miRNA) and the gene expression levels of ASF1B, CD82B, CRISP3, FN1, MEF2B, PD-L1, PPARγ, TERT, TIMP1, TOP2A, and TP53 were evaluated via real-time reverse transcription polymerase chain reaction in BM samples from patients with MM exhibiting favorable and unfavorable prognoses. Additionally, the analysis involved the bone marrow samples from patients undergoing examinations for non-cancerous blood diseases (NCBD). The findings indicate a statistically significant increase in the expression levels of miRNA-124, -138, -10a, -126, -143, -146b, -20a, -21, -29b, and let-7a and a decrease in the expression level of miRNA-96 in the MM group compared with NCBD (*p* < 0.05). No statistically significant differences were detected in the expression levels of the selected miRNAs between the unfavorable and favorable prognoses in MM groups. The expression levels of ASF1B, CD82B, and CRISP3 were significantly decreased, while those of FN1, MEF2B, PDL1, PPARγ, and TERT were significantly increased in the MM group compared to the NCBD group (*p* < 0.05). The MM group with a favorable prognosis demonstrated a statistically significant decline in TIMP1 expression and a significant increase in CD82B and CRISP3 expression compared to the MM group with an unfavorable prognosis (*p* < 0.05). From an empirical point of view, we have established that the complex biomarker encompassing the CRISP3/TIMP1 expression ratio holds promise as a prognostic marker in MM. From a fundamental point of view, we have demonstrated that the development of MM is rooted in a cascade of complex molecular pathways, demonstrating the interplay of genetic and epigenetic factors.

## 1. Introduction

Multiple myeloma (MM) is a hematological malignancy characterized by the uncontrolled proliferation of terminally differentiated B-lymphocytes, specifically plasma cells [1,2]. The pathogenesis and progression of MM are intricately linked to genetic alterations, including chromosomal rearrangements, deletions, duplications, and single nucleotide variations. These alterations can exist at both clonal and subclonal levels, contributing to MM progression and resistance to treatment [3,4]. Mutational landscape studies have identified KRAS (20–25%), NRAS (23–25%), TP53 (8–15%), DIS3 (11%), FAM46C (11%), and BRAF (6–15%) as the most frequently mutated genes [5]. Risk stratification models developed for MM have demonstrated that the prognosis for each cytogenetic abnormality is different [4]. The chromosomes exhibiting the highest frequency of deletions are 1p (30%), 6q (33%), 8p (25%), 12p (15%), 13q (59%), 14q (39%), 16q (35%), 17p (7%), 20 (12%), and 22 (18%), with genes of prognostic significance found to be located on 1p (FAF1, CDKN2C), 1q (ANP32E), and 17p (TP53) [6]. Wang and colleagues developed a prognostic panel for MM by analyzing the expression of the HNRNPC, RPLP2, SNRPB, EXOSC8, RARS2, MRPS31, ZC3H6, and DROSHA genes [7].

The pathogenesis of MM is characterized not only by genetic modifications but also by epigenetic ones [8]. MicroRNAs (miRNAs) have been demonstrated to regulate the expression of oncogenes and tumor suppressor genes, exerting notable influence on diverse biological processes within multiple myeloma. These processes encompass proliferation, angiogenesis, and the heterogeneity of the bone marrow microenvironment [8,9]. A meta-analysis study by Xu and colleagues has revealed that elevated miRNA-92a and the decreased expression of miRNAs like miRNA-16 -25, -744, -15a, let-7e, and miRNA-19b were correlated with poor prognosis in MM [10]. A study analyzing miRNA levels in the blood of myeloma patients treated with bortezomib has found that lower levels of miRNA-328 and miRNA-409 were associated with a higher risk of early death [11]. Che and colleagues have identified miRNA-27 as an oncogene, with its increased expression being associated with unfavorable outcomes for MM patients [12]. Additionally, the expression levels of the miRNA-15a, miRNA-16-1, and miRNA-17-92 clusters have been correlated with poor prognosis in MM [13].

MM exhibits the most extensively characterized prognostic factors, which can be categorized as follows: patient physical status, tumor burden, and inherent cellular characteristics [14]. The immune system plays a crucial role in the development and progression of MM. Studies have shown that immune cells and cytokines like IL-1, IL-2, IL-6, IL-8, IL-12, IL-15, IL-17, IL-18, and IL-32 within the tumor microenvironment can significantly influence the progression of MM in patients. The prognostic value of analyzing alterations in these factors has been confirmed by clinical studies, allowing for MM patient response monitoring and disease progression prediction [15].

Until recently, the overall survival prognosis and the choice of therapeutic tactics in MM patients have been based on scales assessing the general physical condition of the patient. These include the Karnofsky scale (KPS) and ECOG, as well as the Durie–Salmon classification, replaced by the International Staging System and its revised version (ISS and R-ISS), based on a combination of widely available and important prognostic factors in MM related to the disease and tumor mass volume (β2-microglobulin concentration, albumin, hemoglobin, calcium, cytogenetic abnormalities, and others). These two classifications are currently considered the standard model of risk stratification for patients with newly diagnosed multiple myeloma [16]. However, the development of new drugs in the last two decades has made it necessary to search for novel reliable prognostic biomarkers that can be used as indicators of survival and the efficacy of therapy in MM patients [17,18].

Both genetic and epigenetic factors show promise as biomarkers for diagnosing and predicting the course of multiple myeloma, playing a significant role in tumor behavior. Despite numerous studies exploring the impact of genetics and epigenetics on multiple myeloma, effective diagnostic markers for routine clinical use remain limited.

Due to underlying molecular variation, the clinical course of multiple myeloma varies widely among patients. Although some patients experience long periods of remission, others experience early recurrence or their disease is resistant to treatment. To further improve the survival of patients with multiple myeloma, it is necessary to add the information on molecular biomarkers that cause differences in treatment outcomes into prognostic tools that are used to stratify patients into treatment groups.

The aim of this study was to identify biomarkers that can improve the accuracy of prognosis assessment in MM.

## 2. Results

### 2.1. Epigenetic Biomarkers in Multiple Myeloma

A total of 243 miRNAs differentially expressed between MM groups compared to patients with non-cancerous blood diseases (NCBD) were identified by sequencing in bone marrow (BM) samples (Figure 1).

For validation, miRNA-106b, -10a, -126, -143, -145, -146b, -15a, -18a, -197, -20a, -21, -221, -23a, -30b, -31, -551b, -128, -138, -150, -196b, -223, -124, -181a, -26a, -29b, -96, and let-7a were chosen for a comparative analysis of miRNA expression levels utilizing real-time RT-PCR. This analysis was performed in MM groups with distinct prognoses (favorable (n = 28) and unfavorable (n = 17)) and compared to NCBD (n = 43). A statistically significant elevation in the expression levels of miRNA-124, -138, -10a, -126, -143, -146b, -20a, -21, -29b, and let-7a, along with a decrease in the expression level of miRNA-96, was observed in the MM group compared to the NCBD group (*p* < 0.05) (Table 1).

No statistically significant differences in the expression levels of selected miRNAs were observed between the MM groups with poor and good prognoses (*p*-values were in the range from 0.11 to 0.97) (Appendix A).

### 2.2. Genetic Biomarkers in Multiple Myeloma

Based on the literature data, we selected 11 putative biomarkers for MM: ASF1B, CD82B, CRISP3, FN1, MEF2B, PD-L1, PPARγ, TERT, TIMP1, TOP2A, and TP53. These genes are oncogenes and suppressor genes of different tumor types [19,20,21,22,23,24,25,26,27,28,29]. We wanted to determine whether they have aberrant expression in multiple myeloma, too. The presence of variability in their expression in multiple myeloma may mean that they are potential biomarkers of MM. We examined the gene expression levels of ASF1B, CD82B, CRISP3, FN1, MEF2B, PD-L1, PPARγ, TERT, TIMP1, TOP2A, TP53 in the MM samples with contrasting prognostic profiles (favorable (n = 28) vs. unfavorable (n = 17)). These levels were compared to those in the NCBD samples (n = 43). A statistically significant reduction in ASF1B, CD82B, and CRISP3 gene expression was observed in the MM group compared to the NCBD group (*p* < 0.05) alongside a concurrent elevation in FN1, MEF2B, PD-L1, PPARγ, and TERT expression levels (Appendix A). The most substantial differences (of more than threefold) were observed for FN1, PD-L1, PPARγ, and TERT (Figure 2).

A statistically significant reduction in TIMP1 expression was observed, accompanied by an increase in CD82B and CRISP3 expression, in the myeloma (MM) group with a favorable prognosis compared to the MM group with an unfavorable prognosis (*p* < 0.05) (Appendix A and Figure 3). A gene exhibiting a more than twofold difference in expression could be a potential biomarker.

Next, we conducted a ROC analysis on CRISP3, TIMP1, and a complex marker representing the ratio of CRISP3 to TIMP1 (Figure 4). The CRISP3/TIMP1 complex biomarker demonstrated the optimal AUC value, accompanied by high sensitivity and specificity.

### 2.3. Bioinformatics Analysis of MiRNA–MRNA Interaction

The next step involved a bioinformatics analysis of miRNA–mRNA interactions between the miRNAs and the genes under study. The data obtained are presented using the miRNet 2.0 resource (Figure 5). MiRNet is a visual analytics platform designed to explore miRNA networks (https://www.mirnet.ca/miRNet/home.xhtml, accessed on 1 October 2024). The analysis has demonstrated a decreased level of miRNA-96 expression and a corresponding increase in FN1 expression in MM samples. These findings are consistent with the results of bioinformatics analysis, as FN1 is a target of miRNA-96. Additionally, we have found CRISP3 to be a target of miRNA-21. Our observations of elevated miRNA-21 expression and reduced CRISP3 expression are also consistent with bioinformatics analysis results. Furthermore, our analysis has revealed decreased expression of ASF1B alongside increased expression of miRNAs, including miR-126, miR-143, miR-21, miR-29b, and let-7a, which act as its direct regulators. The bioinformatic analysis showed that TERT mRNA is a target of miRNA-143 and miRNA-138. We demonstrated that in multiple myeloma, both miRNA-143 and miRNA-138 expression and the level of TERT are increased. Perhaps there are other mechanisms (underlying the regulation of TERT) that more strongly control changes in its expression than these miRNAs do. This finding indicates a potential regulatory mechanism involving these miRNAs in TERT expression.

## 3. Discussion

We have analyzed the expression levels of the ASF1B, CD82B, CRISP3, FN1, MEF2B, PD-L1, PPARγ, TERT, TIMP1, TOP2A, and TP53 genes in bone marrow samples collected from MM patients with favorable and poor prognoses as well as patients being examined for NCBD. The ASF1B, CD82B, and CRISP3 expression levels were found to be statistically significantly downregulated, while the expression levels of FN1, MEF2B, PD-L1, PPARγ, and TERT were increased in MM patients compared to the NCBD group; the complex biomarker (the CRISP3/TIMP1 expression ratio) was shown to be a potential prognostic biomarker in MM.

Numerous studies have demonstrated that anti-silencing function 1B histone chaperone (ASF1B) is a key regulator of proliferation, apoptosis, and the cell cycle in different types of cancer and that its expression correlates with the clinical data and with the overall survival rates in particular. Zhang et al. pooled the results of the studies focusing on 26 different types of tumors and showed ASF1B overexpression in 25 types of cancer cells in solid tumors and its decreased level in acute myeloid leukemia [19]. We observe decreased ASF1B levels in MM (*p* < 0.05). It is worth noting that the reduced ASF1B level may be indicative of the features of MM cells and hematologic malignancies in general.

The genes belonging to the myocyte enhancer factor 2 (MEF2) family (MEF2-A, -B, -C, and -D) were characterized as oncogenes in hematologic malignancies. MEF2B was reported to play a unique role compared to its paralogs in B cells [20]. We observe that the MEF2B expression level in MM is increased 2.5-fold (*p* < 0.05). MM is a B-cell malignancy [30]. Hence, our findings emphasize the association between MEF2B and B cells.

PD-L1 also plays a crucial role in different malignancies where it can attenuate the host immune response to tumor cells [31]. In different cancers, PD-L1 overexpression is associated with poor prognosis [21]. Tamura et al. demonstrated that plasma cells obtained from MM patients expressed higher PD-L1 levels compared to the control group and patients with monoclonal gammopathy of undetermined significance [32]. We observe that the PD-L1 levels in MM patients increased almost fourfold (*p* < 0.05), which is consistent with the data published previously.

Peroxisome proliferator-activated receptors (PPARs) are ligand-dependent transcription factors belonging to the nuclear hormone receptor superfamily. PPARγ was shown to act as a tumor suppressor promoting proliferation inhibition in colon and breast cancer [22]. In pancreatic cancer cells, PPARγ ligands induced apoptosis and growth inhibition linked to the G1-phase arrest of cell cycle progression through the overexpression of the p27Kip1 protein [33]. Garcia-Bates et al. showed that overexpression of PPARγ promotes apoptosis in MM cells [34]. We observe that the PPARγ level in MM patients is increased more than fourfold (*p* < 0.05). In the present study, the trend toward the upregulation of PPARγ, which was previously reported to act as a tumor suppressor, is related to the clinical features of the analyzed sample. Hence, one of the promising objectives is to determine whether PPARγ expression levels are changed as MM progresses.

Fibronectin 1 (FN1) mediates the interplay between cells and the extracellular matrix; it plays an important role in cell adhesion, migration, growth, and differentiation. FN1 was shown to promote cell proliferation and migration in stomach cancer cell lines [23]. Upregulated FN1 expression correlates with poor prognosis in breast cancer [35]. FN1 expression was increased in different subtypes of thyroid carcinoma; decreased FN1 levels were associated with the suppression of cell proliferation and invasion [36]. FN1 overexpression is associated with poor survival in squamous cell carcinoma of the esophagus [37]. FN1 expression is upregulated in MM and can be used as a potential biomarker in MM diagnosis [38]. We observe a more than fourfold upregulation of FN1 levels in MM patients (*p* < 0.05), which is consistent with the data published previously.

Telomerase reverse transcriptase (TERT) is the telomerase catalytic subunit. Research demonstrates that TERT plays a crucial role in cancer development, and its expression is regulated by different miRNAs [24]. It was reported that TERT overexpression in tumors is a prognostic factor for poor prognosis [39]. Aref et al. showed that the overall survival of MM patients correlated with telomere length and the TERT genotype [40]. We observe that TERT expression is upregulated more than threefold in MM patients (*p* < 0.05), but there is no statistically significant correlation between TERT expression and the assessed prognosis of MM. The aberrant expression of TERT may possibly contribute to the development of MM, but structural changes in it are the factor significantly affecting disease aggressiveness.

KAI1/CD82 is a well-studied metastasis suppressor in different solid malignant neoplasms. KAI1/CD82 expression is downregulated in malignant tumors and is closely associated with malignant progression and metastatic spread and prognosis (including breast, colon, lung, ovarian, nasopharyngeal, liver, and pancreatic cancers) [25,41]. Tohami et al. demonstrated that CD81/CD82 have a negative effect on the adhesion, motility, and invasion of MM cells, thus confirming their role as metastasis suppressors [42]. We observe downregulated CD82 expression in MM cells compared to NCBD patients (*p* < 0.05), which is consistent with the data published previously. Meanwhile, we observe upregulated CD82 expression in MM cells with favorable prognosis vs. poor prognosis (*p* < 0.05), which confirms that CD82 acts as a tumor suppressor.

Tissue inhibitors of metalloproteinases (TIMPs), important regulators of hydrolysis or the activation of matrix metalloproteinases (MMPs), are involved in the emergence and development of MM [43]. TIMP was found to regulate such processes as angiogenesis, apoptosis, and cellular differentiation [44]. TIMP1 was reported to function as an oncogene in different types of cancer. Aberrant TIMP1 overexpression in cancer-associated fibroblasts stimulates tumor progression through CD63 in lung adenocarcinoma [26]. TIMP1 is a prognostic marker of the progression and metastatic spread of colorectal cancer via the FAK-PI3K/AKT and MAPK pathways. The suppression of TIMP1 expression reduces proliferation and metastatic spread, as well as intensifying apoptosis [45]. The overall survival and post-progression survival were significantly lower in MM patients with high levels of TIMP1 protein [46]. High serum levels of TIMP1 correlate with advanced disease and are a predictor of poor survival in MM patients [47]. We observe that TIMP1 expression is statistically significantly downregulated in samples from MM patients with a favorable prognosis (*p* < 0.05), which is consistent with the data published previously, confirming the feasibility of using TIMP1 as a prognostic biomarker in MM.

Cysteine-rich secretory protein 3 (CRISP3) encodes the extracellular matrix protein; it was reported to be significantly downregulated in cervical cancer, especially in cases with HPV16 infection, which was associated with poorer overall survival [27]. Ko et al. noted that CRISP3 acts as a suppressor in oral squamous cell carcinoma [48]. It was reported that CRISP3 significantly upregulated in MM is a potential peripheral blood biomarker in MM [49]. We observe decreased CRISP3 levels in bone marrow specimens from MM patients compared to the NCBD group (*p* < 0.05), as well as downregulated expression in the MM group with poor prognosis (*p* < 0.05), which further characterizes it as a potential tumor suppressor.

We performed miRNA sequencing in bone marrow specimens, followed by validation of the results for the analyzed groups that comprised a sufficient number of samples to obtain statistically significant results. It was revealed that the expression of miRNA-124, -138, -10a, -126, -143, -146b, -20a, -21, -29 b, and let-7a was statistically significantly upregulated, while the expression of miRNA-96 was downregulated in MM patients compared to the NCBD group (*p* < 0.05).

Bioinformatics analysis revealed that oncogenes and tumor suppressor genes selected by us to identify their roles in MM are involved in complex molecular pathways and simultaneously controlled by several miRNAs, which were found to be expressed in MM by sequencing. It is worth noting that some of these miRNA–mRNA interactions have already been proven experimentally in different diseases.

The literature review demonstrated that the expression level of miRNA-138 varies depending on the type of tumor being analyzed [50]. Previously, Yan et al. showed that miRNA-138 expression is upregulated in MM and regulates proliferation by targeting PAX5 [51]. In this study, we also observe the upregulated miRNA-138 expression in MM (*p* < 0.05). Thus, miRNA-138 expression is downregulated in thyroid cancer, which promotes TERT overexpression and correlates with the cancer stage and the invasive phenotype [52]. MiRNA-138 suppresses proliferation, migration, and invasion by targeting TERT in cervical cancer [53]. Yan et al. demonstrated that TIMP1 is targeted by miR-138 in osteoporosis [54]. Rostami et al. showed that PDL1 is targeted by miRNA-138 in non-small-cell lung cancer [55].

Yuan et al. demonstrated that miRNA-20a functions as an oncogene in MM and enhances the proliferation and migration of plasma cells, as well as inhibiting apoptosis through the regulation of PTEN [56]. We observe the overexpression of miRNA-20a in MM, which is consistent with its previously reported function as an oncogene in MM.

Chen et al. showed that miRNA-10a promotes cancer cell proliferation in oral squamous cell carcinoma through the upregulation of GLUT1 expression and the stimulation of glucose metabolism [57]. The overexpression of miRNA-10a promotes tumor progression in cervical cancer by suppressing UBE2I signaling [58]. MiRNA-10a promotes the development of non-small-cell lung cancer by regulating the expression of PTEN tumor suppressor [59]. miRNA-10a suppresses breast cancer progression via the PI3K/Akt/mTOR pathway [60]. miR-10a suppresses cell proliferation and promotes apoptosis by targeting BCL6 in diffuse large B-cell lymphoma [61].

A meta-analysis of 17 articles revealed that miR-126 overexpression is a favorable prognostic factor for overall survival in different cancers [62]. It was mentioned that miR-126 may play an accessory role in cancer progression due to the mediated stimulation of vascular growth [63]. Thus, epithelial ovarian cancer patients with miR-126 overexpression had poor overall survival and relapse-free survival [64]. Liu et al. showed that miRNA-126 induced the apoptosis of myeloma cells by reducing the level of anti-apoptotic protein MCL [65].

Long-term analysis of the roles played by miR-143 and -145 in tumors of different localization demonstrated that they can have both pro-oncogenic and anti-oncogenic functions depending on the type of tissue analyzed [66]. Gupta et al. showed that the relative expression of miR-143, miR-144, miR-199a, and miR-203 in both blood and bone marrow is downregulated in MM patients compared to the control group [67]. However, we observe statistically significant upregulation of miRNA-143 in MM patients (*p* < 0.05), which may be due to both differences in the clinical characteristics of the specimens in the studies and technical differences during the analysis of expression levels.

Shi et al. showed that miR-146b promotes the progression of colorectal cancer by targeting TRAF6 [68]. miR-146b functions as a suppressor in non-small-cell lung cancer and can inhibit cell proliferation, clonogenicity, and migration, as well as regulate the cell cycle [69]. These data represent the general trend for many miRNAs that they can function both as an oncogene and as a suppressor depending on the type of tissue in which they are expressed. Low-level expression of miRNA-146b predicts unfavorable outcomes of treating large B-cell lymphoma using cyclophosphamide, doxorubicin, vincristine, and prednisone [70]. MicroRNA-146b overexpression is associated with clinical worsening, the more advanced ISS stage, chromosome abnormalities, and poor prognosis in MM patients [71]. However, in our study, we obtained no statistically validated data indicating that miRNA-146b expression level varies according to the aggressiveness of MM. Li et al. showed that miRNA-146b can mediate PPARγ regulation through SIAH2 [72]. We observe statistically significant upregulation of both miRNA-146b and PPARγ in MM, which may support the fact that PPARγ regulation is mediated by miRNA-146b.

Wang et al. demonstrated that miR-20a promotes proliferation and inhibits the apoptosis of MM cells in vitro through EGR2 regulation [73]. Jiang et al. showed that miR-20a inhibition suppresses the progression of MM by modulating the PTEN/PI3K/Akt signaling pathway [74]. The upregulated expression of miR-20a promotes osteosarcoma progression by directly targeting QKI2 [75]. miRNA-20a promotes the proliferation of non-small-cell lung cancer cells by upregulating PD-L1 through the mediated targeting of PTEN [76]. We observe statistically significant upregulation of miRNA-20a and PD-L1 in MM, which possibly occurs in a manner similar to the previously described pathway through mediated PTEN regulation. Further studies are needed to confirm this hypothesis.

miR-21 is characterized by upregulated expression in most cancers, including MM, and functions as an oncogene by decreasing PTEN expression [77]. Drevytska et al. showed that changes in telomerase activity can significantly alter miRNA levels; in particular, decreased TERT levels promoted the downregulation of miRNA-21, miRNA-29a, and miRNA-208a [78]. We observe miRNA-21 overexpression and the simultaneous upregulation of TERT (*p* < 0.05) in MM, which possibly supports the statement about the mutual regulation of these participants in complex biological pathways that was published earlier.

MiRNA-124 was reported to function as a suppressor in many types of cancer [79]. Sabour Takanlu et al. demonstrated that the upregulation of miRNA-124 and the corresponding downregulation of the expression of its target, EZH2, reduce the proliferation of MM cells [80]. Roshani Asl et al. showed that miRNA-124 suppresses PD-L1 expression and has a suppressive effect in colorectal cancer through the modulation of STAT3 signaling [81]. They also demonstrated that the increased expression of PPARγ suppresses the production of pro-inflammatory cytokines by upregulating miR-124 both in vitro and in vivo [82]. We observe statistically significant upregulation of miRNA-124 and PPARγ in MM (*p* < 0.05), indicating that this regulatory pathway can also exist in MM cells.

Wang et al. showed that miR-29b expression was downregulated in cell lines and tissues of MM; miR-29b slowed down MM progression by decreasing FOXP1 expression [83]. Amodio et al. demonstrated that miR-29b regulates SOCS-1 expression through promoter demethylation and negatively regulates the migration of MM cells [84]. We observe that miR-29b expression in MM patients non-significantly increased less than 1.5-fold (*p* < 0.05). Our findings are not consistent with the previously published data, which may be caused by technical differences in study conduct.

Functioning as oncogenes or tumor suppressors, miRNAs regulate signaling pathways associated with the development and progression of MM. It is important to emphasize that just like one gene is regulated by multiple miRNAs, one miRNA can regulate dozens of target genes. Therefore, the aberrant expression of each miRNA may contribute to a cascade of molecular alterations, which poses difficulties in analyzing the foundations of the pathogenesis of MM. Furthermore, there still exist no standardized protocols for specimen collection, small RNA isolation, and data analysis methods for miRNA quantification.

In general, although searching for prognostic markers in MM is a relevant and urgent problem of molecular biology, the data obtained have not been integrated into routine clinical practice yet. The identification of markers of multiple myeloma progression is the first step in the search for potential targets for the development of new ways to reduce tumor cell clones. Multi-site studies involving large samples of analyzed specimens need to be conducted to validate the data obtained. The study of a larger number of molecular biomarkers involved in the pathogenesis of MM will allow one to elaborate a better diagnostic system and personalize therapy.

## 4. Materials and Methods

### 4.1. Clinical Samples

A total of 88 cytological samples were obtained by sternal puncture and aspiration biopsy of BM from the posterior iliac spine. The study groups included MM (n = 45); the control group consisted of NCBDs (n = 43) (iron-deficiency anemia (n = 28), hemolytic anemia (n = 5), B12 deficiency anemia (n = 5), and immune thrombocytopenia (n = 5)). All the cases were patients before treatment initiation.

The patients were distributed into the groups of favorable (n = 28) and unfavorable (n = 17) prognoses according to the International Staging System (ISS) at first diagnosis of the disease. According to this staging system, the patients analyzed in our study and having stage I according to the ISS were assigned to the group of favorable prognosis (a predicted median of overall survival of 62 months), whereas patients with stage II or III according to ISS were assigned to the group of unfavorable prognosis (a predicted median overall survival of 44 and 29 months, respectively).

All patients included in the study received standard first-line chemotherapy cycles based on bortezomib within protocols PAD or VCD (for patients under 65 years of age) or within a combination of VMP with VCD (for multiple myeloma patients over 65 years of age) according to clinical guidelines for the diagnosis and treatment of multiple myeloma.

Appendix A summarizes group characteristics. Specimens were obtained in compliance with Russian laws and regulations; each patient had provided written informed consent; and all the data were anonymized. Material collection for this study started on 10 January 2022 and ended on 1 May 2024. The study was conducted in accordance with the Declaration of Helsinki and approved by the Ethics Committee of Novosibirsk State Medical University, study protocol No. 15 of 25 May 2020.

### 4.2. Isolation of Total RNA from Fine-Needle Aspiration Cytological Specimens

Each dried cytological specimen was washed in a microcentrifuge tube with three 200 µL portions of guanidine lysis buffer. Samples were vigorously mixed and incubated in a thermal shaker at 65 °C for 15 min. Next, an equal volume of isopropanol was added. The solution was thoroughly mixed and kept at room temperature for 5 min. After centrifugation at 14,000× *g* for 10 min, the supernatant was decanted, and the pellet was washed with 500 µL of 70% ethanol and 300 µL of acetone. The resulting RNA was dissolved in 290 µL of deionized water.

### 4.3. RNA Sequencing

RNA was used to obtain libraries for massive parallel sequencing using a NEB-Next^®^ Small RNA Library Prep Set for Illumina^®^ E7330S (NEB) according to the kit instructions. Library concentration was determined using a Qubit dsDNA HS Assay Kit (Thermo Fisher Scientific, Waltham, MA, USA) on a Qubit 2.0 fluorometer. Fragment length distribution was conducted using an Agilent High Sensitivity DNA Kit (Agilent, Santa Clara, CA, USA). Sequencing was performed on a HiSeq1500 (Illumina, San Diego, CA, USA) with a generation of at least 5 × 106 short 50 nt reads.

Adapters sequences were removed from reads using cutadapt v. 3.1; the reads were then mapped against the human genome (GRCh38) using bowtie version 1.0.1. microRNA quantification in the resulting alignments was performed using the feature Counts v. 1.5.0-p1 software with minOverlap 14-Q 10 parameters for the mirbase v. 22 database.

### 4.4. Validation of RNA Sequencing Results by RT-qPCR Analysis

The sequences of all oligos are listed in Appendix A. The geometric mean of miR-378, miR-103a, and miR-191 expression levels was used for normalization [85,86]. Oligonucleotides were selected using the PrimerQuest online service (https://eu.idtdna.com/, accessed on 10 January 2019). The qPCR threshold cycles were analyzed using the 2^−ΔCt^ method [87].

### 4.5. Gene Expression Analysis by RT-qPCR Method

Based on the literature data, we selected 11 putative biomarkers for MM: ASF1B, CD82B, CRISP3, FN1, MEF2B, PD-L1, PPARγ, TERT, TIMP1, TOP2A, and TP53. These genes are oncogenes and suppressor genes of different tumor types [19,20,21,22,23,24,25,26,27,28,29]. Phosphoglycerate kinase 1 (PGK 1) was selected as a reference gene [88]. The threshold cycles obtained by qPCR were analyzed using the 2^−ΔCt^ method.

### 4.6. Reverse Transcription

The reverse transcription reaction for cDNA was conducted in a volume of 30 µL. Ready-to-use RealBest RT Master Mix (Vector-Best, Novosibirsk, Russia) was utilized. The reverse transcription reaction contained 3 μL of RNA preparation, 21.6% trehalose, 1× RT buffer (Vector-Best, Novosibirsk, Russia), 0.4 mM of each dNTP, 1% BSA, 100U M-MLV reverse transcriptase (Vector-Best, Novosibirsk, Russia), and 0.2 μM of appropriate RT primer. All oligonucleotides were synthesized by Vector-Best (Novosibirsk, Russia). The cDNA-containing reaction mixture (3 µL) was used immediately as a template for real-time PCR on a CFX96 system (Bio-Rad, Hercules, CA, USA).

### 4.7. Real-Time PCR

The miRNA expression levels were measured by real-time PCR on a CFX96 amplifier (Bio-Rad Laboratories, Hercules, CA, USA). The total volume of each reaction was 30 μL; the reaction mixture contained 3 μL of cDNA, 1× PCR buffer (Vector-Best, Novosibirsk, Russia), 0.4 mM of each dNTP (Biosan, Riga, Latvia), 1% BSA, 1U Taq polymerase (Vec-tor-Best, Novosibirsk, Russia) premixed with 10× active site-specific monoclonal antibody (Clontech, Mountain View, CA, USA), 0.5 units of uracil-DNA glycosylase (Vector-Best, Novosibirsk, Russia), 0.5 μM of each primer, and 0.25 μM of TaqMan probe. The primers and the probes were developed by Vector-Best; the PCR efficiency was 90–100%.

### 4.8. Statistical Analysis

Statistical analysis was performed using the Statistica v13.1 software. The Mann–Whitney U test was used; *p* values < 0.05 were considered statistically significant.

The diagnostic values were evaluated by the receiver operating characteristic (ROC) curves analysis (IBM SPSS software platform, version 26).

Bioinformatics analysis of miRNA target genes was conducted using the miRNet 2.0 tool (https://www.mirnet.ca/miRNet/home.xhtml, accessed on 1 October 2024).

## Figures and Tables

**Figure 1 ijms-25-13404-f001:**
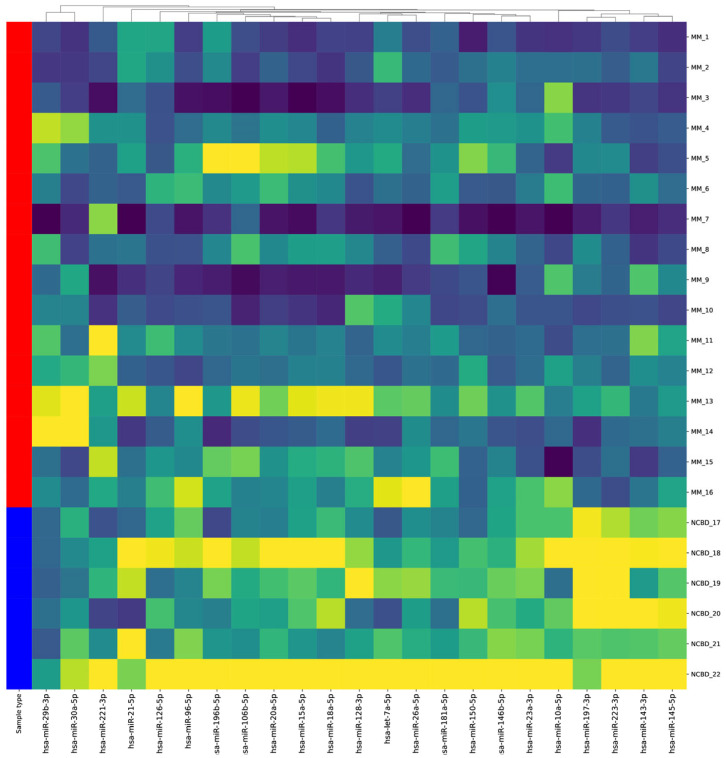
Hierarchical cluster analysis between 16 multiple myeloma (MM) cases and six non-cancerous blood diseases (NCBD) cases for the microRNAs (miRNAs) that were chosen for validation by RT-PCR in the analyzed groups. Each column represents the expression of a miRNA, and each row denotes a nucleic acid sample. Yellow: upregulated miRNA; blue: downregulated miRNA; green: minor changes; red: a graphical representation of a group of samples.

**Figure 2 ijms-25-13404-f002:**
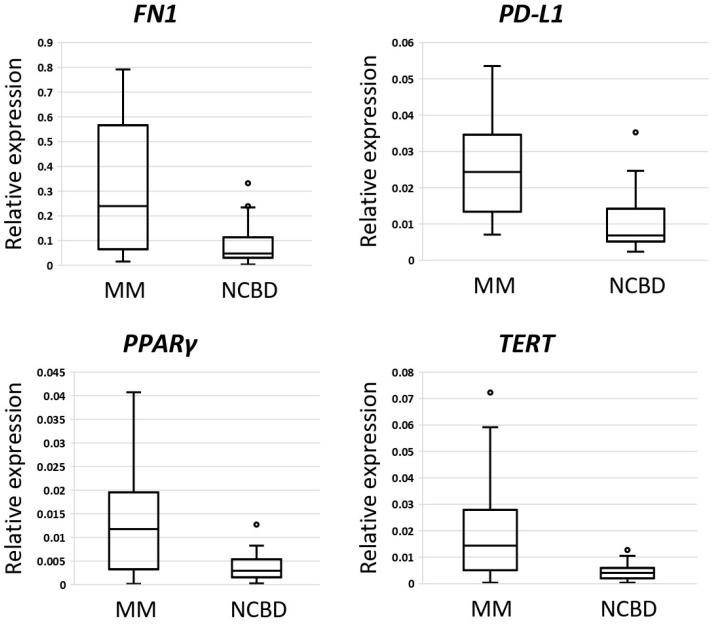
Comparative analysis of gene expression levels between multiple myeloma (MM) (n = 45) and non-cancerous samples (NCBD) (n = 43). The figure presents the median value, upper and lower quartiles, non-outlier range, and outliers appearing as circles.

**Figure 3 ijms-25-13404-f003:**
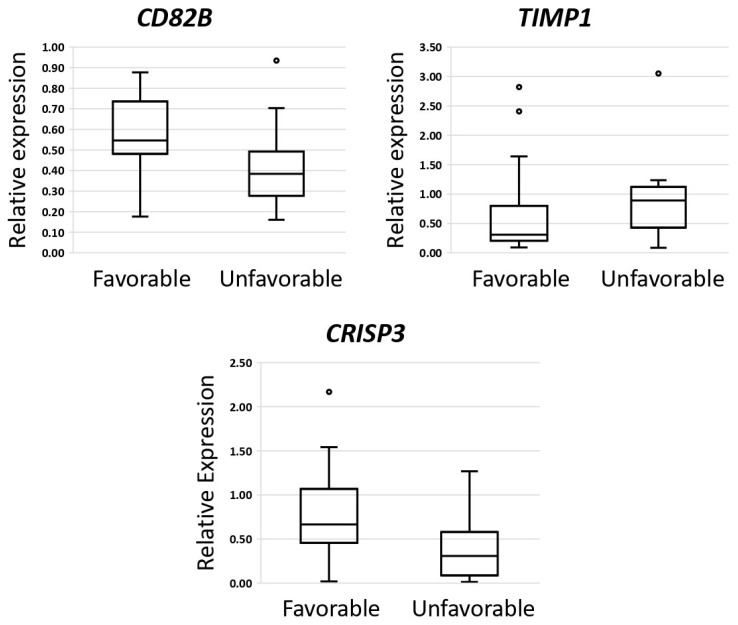
Comparative analysis of gene expression levels between multiple myeloma samples of patients with favorable (n = 28) and unfavorable (n = 17) prognosis. The figure presents the median value, upper and lower quartiles, non-outlier range, and outliers appearing as circles.

**Figure 4 ijms-25-13404-f004:**
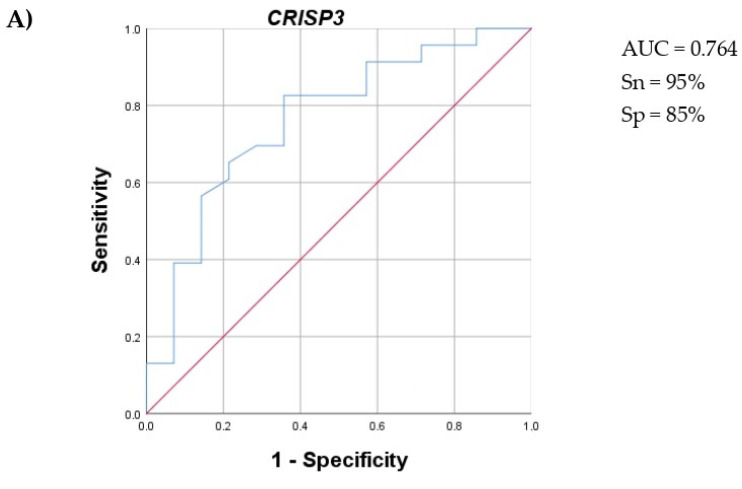
ROC analysis for the (**A**) *CRISP3*, (**B**) *TIMP1*, and (**C**) *CRISP3*/*TIMP1* genes. AUC, sensitivity (Sn), and specificity (Sp) values are indicated. Red line is a diagonal support line, blue is a ROC curve.

**Figure 5 ijms-25-13404-f005:**
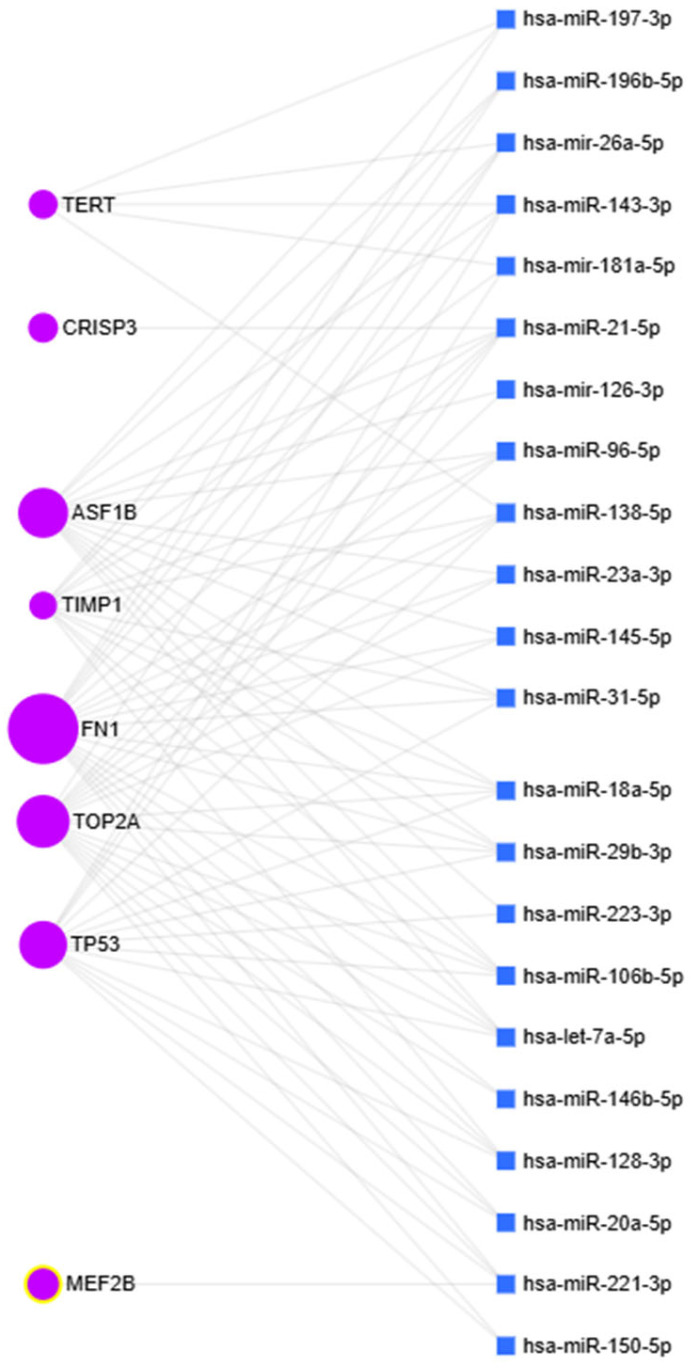
Interactions between microRNAs(miRNAs) and their target genes. Blue squares represent miRNAs, and purple circles indicate their target genes.

**Table 1 ijms-25-13404-t001:** Comparative analysis of microRNA expression levels between multiple myeloma (n = 45) and non-cancerous samples (n = 43).

	Fold Change	*p*-Value
miR-96	−2.05	1 × 10^−6^
miR-124	3.54	1 × 10^−6^
miR-138	11.1	1 × 10^−5^
miR-10a	3.26	1 × 10^−2^
miR-126	1.47	1 × 10^−3^
miR-143	1.91	3 × 10^−2^
miR-146b	2.03	2 × 10^−4^
miR-20a	1.39	3 × 10^−2^
miR-21	1.53	1 × 10^−3^
miR-29b	1.45	1 × 10^−4^
Let-7a	6.99	1 × 10^−15^

## Data Availability

All data generated or analyzed during this study is available from the corresponding author and will be provided on request.

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
