# Peer review of "Multiple Myeloma: Genetic and Epigenetic Biomarkers with Clinical Potential"

_ijms, 2024, doi:10.3390/ijms252413404_

Round 1

Reviewer 1 Report

Comments and Suggestions for Authors

The manuscript entitled: “Multiple myeloma: genetic and epigenetic biomarkers with clinical potential” by Veryaskina et al. aims to identify biomarkers with the potency to improve the accuracy and prognosis in MM disease.

Albeit the paper is well written and of interest, comments should be addressed to further improve the manuscript.

Comments:

1.    Introduction page 2 line 91-92 and 97: Please provide more information about the rational of this study. Moreover, please clarify how the accuracy of the prognosis in MM patients is lacking.

2.    Results page 3, line 112-113: please demonstrate these results also clearly.

3.    Results page 5 Figure 1. The AUC of Figure A A) and B) is not quite good, so the conclusions should be drawn carefully.

4.    Discussion section: is much too long and should be shortened. Moreover, the discussion section should be more balanced according to advantages and weakness of the study. Due to the rather small cohort of MM patients, the conclusions should be balanced. Moreover, provide some more information about the current consequences which the clinicians should be drawn from this study. How could these results add more information in comparison to the current prognostic features.  Moreover, how could these biomarkers improve the current accuracy of prognosis assessments in MM patients.

Reviewer 2 Report

Comments and Suggestions for Authors

I have several comments:

1. The authors mentioned that 243 miRNA were differentially expressed after sequencing - I suppose they meant NGS. Please, add the heat map and hierarchical clustering of miRNA.

2. Aspirational biopsy of the BM usually yields plasma cells that need to be separated by CD138+ marker. Was this separation performed? 

3. For sternal puncture, did the authors check for plasma cell purity?

4. Please add clinical characteristics of MM patients - infiltration of bone marrow, ISS stage. Please confirm that patients were tested at diagnosis. If not please add treatment.

5. The control group seems to be quite heterogeneous; plus, all the patients are sick - perhaps they are not a good control. Why didn't the authors use the good prognosis vs bad prognosis MM groups of patients? That would make more sense.

6. why did the authors use geometrical means of 3 microRNAs as endogenous control (line 413)?

7. It seems unclear why the authors chose the 11 genes to test. Please explain.

8. How many samples were used for the ROC analysis? Were some of these samples also sequenced?

9. How was poor and good prognosis of MM patients defined? How were the patients treated?

Reviewer 3 Report

Comments and Suggestions for Authors

The manuscript "Multiple myeloma: genetic and epigenetic biomarkers with clinical potential", written by Veryaskina YA, Titov SE, Skvortsova NV, Kovynev IB, Antonenko OV, Demakov SA, Demenkov PS, Pospelova TI and Zhimulev IF. analyzes the expression of differentially expressed genes and miRNA in the bone marrow samples from multiple myeloma (MM) patients in comparison with expression in samples from patients with noncancerous blood diseases as control samples. The aim was to identify biomarkers which could better predict prognosis in MM.The authors analyzed expression of 27 miRNA and found differences in expression of 11 miRNA between control and MM samples, but not between samples with favorite and poor prognosis. Also, differential expression was shown for 8 genes, among which, expression of TIMP1 (metalloprotease inhibitor) and CD82B and CRISP3 were shown to differ between patients with favorable and unfavorable prognosis, and the ratio between CRISP3 and TIMP1 expression was suggested as possible prognostic marker.

The Introduction presents the basic data about multiple myeloma, but some facts related to the pathogenesis of MM are missing (such as phases of development, characteristics of the cells – such as production of monoclonal antibodies). There is a detailed list of mutations and chromosome deletions found in MM, as well as miRNA linked with MM diagnosis and poor prognosis.

The Results start with results of miRNA analysis. As Materials and methods are at the end of the article, it seems to me that it should be explained first what was done, which samples were taken, how many samples were analyzed, what was the procedure with samples, even supplement table with data about patients could be put in the results. miRNA expression analysis is given in the form of table, and it could be presented in the form of histogram or other graphical form. Also, basic criteria for "poor and good prognoses" can be explained (possibly in the Materials and methods) and results presented in the form of histogram. The second paragraph describes differences in the gene expression between samples. It could be explained how these genes were chosen (it is in Materials and methods), how many samples were favorable and unfavorable, and data could also be presented in a graphical form. In all tables/graphs it should be mentioned the number of samples and methods used. In Discussion, a description of each of differentially expressed genes is presented. For most of the genes its full names and roles are given for the first time in Discussion, and their names could be mentioned already in the Results. References are missing in lines 233-234, or explanation of the mechanism by which would structural changes in TERT affect the aggressiveness of the disease – TERT expression by itself is the factor of tumorigenesis as it is expressed in tumor and stem cells.

In Materials and methods the criteria for dividing MM groups with favorable and unfavorable diagnosis are missing. Also, the abbreviation PGK should be explained.

Other comments

line 165: correction needed

Round 2

Reviewer 1 Report

Comments and Suggestions for Authors

The manuscript entitled: “Multiple myeloma: genetic and epigenetic biomarkers with clinical potential” by Veryaskina et al. aims to identify biomarkers with the potency to improve the accuracy and prognosis in MM disease.

After revision of the manuscript, the authors addressed all my comments sufficiently.

Reviewer 2 Report

Comments and Suggestions for Authors

My comments have been answered.